# Superpixel-Based Singular Spectrum Analysis for Effective Spatial-Spectral Feature Extraction

Subhashree Subudhi [1][ID], Ramnarayan Patro [1][ID], Pradyut Kumar Biswal [1,*] and Fabio Dell'Acqua [2][ID]

1 Department of Electronics and Communication Engineering, IIIT, Bhubaneswar 751003, India; C116013@iiit-bh.ac.in (S.S.); C116009@iiit-bh.ac.in (R.P.)
2 Department of Electrical, Computer and Biomedical Engineering, University of Pavia, 27100 Pavia, Italy; fabio.dellacqua@unipv.it
* Correspondence: pradyut@iiit-bh.ac.in

**Abstract:** In the processing of remotely sensed data, classification may be preceded by feature extraction, which helps in making the most informative parts of the data emerge. Effective feature extraction may boost the efficiency and accuracy of the following classification, and hence various methods have been proposed to perform it. Recently, Singular Spectrum Analysis (SSA) and its 2-D variation (2D-SSA) have emerged as popular, cutting-edge technologies for effective feature extraction in Hyperspectral Images (HSI). Using 2D-SSA, each band image of an HSI is initially decomposed into various components, and then the image is reconstructed using the most significant eigen-tuples relative to their eigen-values, which represent strong spatial features for the classification task. However, instead of performing reconstruction on the whole image, it may be more effective to apply reconstruction to object-specific spatial regions, which is the proposed objective of this research. As an HSI may cover a large area, multiple objects are generally present within a single scene. Hence, spatial information can be highlighted accurately by specializing the reconstruction based on the local context. The local context may be defined by the so-called superpixels, i.e., finite sets of pixels that constitute a homogeneous set. Each superpixel may undergo tailored reconstruction, with a process expected to perform better than non-spatially-adaptive approaches. In this paper, a Superpixel-based SSA (SP-SSA) method is proposed where the image is first segmented into multiple regions using a superpixel segmentation approach. Next, each segment is individually reconstructed using 2D-SSA. In doing so, the spatial contextual information is preserved, leading to better classifier performance. The performance of the reconstructed features is evaluated using an SVM classifier. Experiments on four popular benchmark datasets reveal that, in terms of the classification accuracy, the proposed approach overperforms the standard SSA technique and various common spatio-spectral classification methods.

**Keywords:** hyperspectral image; superpixel segmentation; evaluation; 2D-singular spectrum analysis (2D-SSA); feature extraction



## 1. Introduction

Recent advancements in hyperspectral sensors resulted in the increased availability of Hyperspectral Images (HSI) and a boost in their circulation among the remote sensing community. HSI data enables the discrimination of objects even with minor differences as it contains several contiguous spectral bands acquired from the visible to the infrared region [1] so that every small spectral difference can, in principle, be captured. The information is available in the form of a 3-D structure that contains a 2-D spatial scene along with a 1-D spectral signature. These unique characteristics of HSI have made them popular in several application areas, such as agriculture [2], mineralogy [3], land cover classification [4], target detection [5], and others. However, effective classification of HSI is still an open challenge.

Several classification techniques, such as K-nearest neighbor (KNN) [6], support vector machine (SVM) [7], multinomial logistic regression (MLR) [8], Extreme Learning Machine (ELM) [9], and Sparse Representation Classifier (SRC) [10] have been proposed in the past decades. The richness in spectral information attracted research efforts on pixel-based processing and classification. SVM is the most popular and widespread classifier due to its lower generalization error rate that makes it capable of identifying even minor changes in spectral signatures. Due to the high spectral dimensionality compared to a generally limited number of class-specific training samples, it is quite difficult to properly estimate the model parameters. Hence, there is a need to adopt effective spatial-spectral feature extraction approaches to overcome the aforementioned challenges.

To deal with the issue of higher spectral dimensionality, several linear (e.g., Principal Component Analysis (PCA) [11], Independent Component Analysis (ICA) [12], and Linear Discriminant Analysis (LDA) [13]) and non-linear (manifold learning [14]) dimensionality reduction (DR) methods have been introduced. Band selection approaches may also be utilized to select the most informative bands out of several available bands [15,16]. Spectral features alone, however, may not be sufficient to score very high accuracy values. To improve performance, it is necessary to incorporate spatio-spectral features that help increasing the separability of classes.

Recently, various spatial feature extraction techniques have been proposed. Mathematical Morphology is one of the most popular approaches that is extensively utilized by researchers. The concept of Extended Morphological Profiles (EMP) for FE in HSI was first proposed by Benediktsson et al. [17]. This technique utilizes morphological opening and closing transformations to extract spatial geometrical information. Later, Dalla Mura et al. [18] proposed Morphological Attribute Filters (MAP) for the spatial FE. From that point onwards, several variations of Attribute Profiles (AP) were created. Ghamisi et al. [19] conducted a comprehensive survey on the evolution in Attribute Profiles.

Texture Descriptors, including Wavelet transform [20], Gray-Level Co-occurrence Matrix (GLCM) [21], Local Binary Patterns (LBP) [22], and Gabor filters [23], have also been used in the literature for spatial FE. Filtering, i.e., moving-window-based processing, is another approach to extract spatial-spectral features. Various edge-preserving filters, such as Bilateral Filters [24], Trilateral Filters [25], Guidance filters [26], and Domain Transform Recursive Filters [27] have been introduced for spatial FE in the literature. The texture and noise variations are minimized by performing smoothing operations; however, important details, such as edges and lines, are well preserved by these filters [28].

In addition to these techniques, 2D-SSA is another interesting approach for spatial feature extraction. Using this approach, each band image of HSI is initially decomposed into varying trends, oscillations, and noise. Later, HSI is reconstructed using the selected oscillations and trends [29,30]. In 2D-SSA, the spatial structural information is extracted by utilizing the characteristics of the surrounding pixels in a specific embedding window. It can withstand high levels of noise and generally achieves good data classification results.

2D-SSA suffers, however, from several limitations, such as reduced utilization of the abundant spectral information available in data, and over-smoothing or under-smoothing of classification results because of the fixed embedding window size. To overcome the challenge of selecting the optimal embedding window size, recently, multi-scale 2D-SSA has been proposed for the effective extraction of discriminative spatial features under different noise conditions [31,32].

Superpixel segmentation techniques have gained popularity in recent years due to their capability of exploiting spatial structural information adaptively in an image. In [33], a survey on superpixel segmentation as a preprocessing step in HSI analysis was presented. A superpixel-based classification via multiple kernels (SCMK) approach was proposed in [34]. In [35], a region-based relaxed multiple kernel (R2MK) method was proposed that combines the multiscale spectral and spatial features using a kernel collaborative representation classification technique.

To obtain superior classification performance and solve the problem of optimal superpixel number selection, an adjacent superpixel-based multiscale spatial-spectral kernel (ASMGSSK) was proposed in [36]. In [37], a multiscale segmentation-based SuperPCA model (MSuperPCA) was developed, which can effectively integrate multiscale spatial information to obtain the optimal classification result by decision fusion.

Recently, deep learning techniques have become quite popular in the classification of HSI data due to their ability to extract discriminant and abstract features by using a series of hierarchical layers. The initial layers usually extract texture and edge information, whereas deeper layers highlight more complicated features. Some of the most popular deep learning frameworks include stacked autoencoders (SAE) [38], Deep Belief Networks (DBN) [39], Convolution Neural Networks (CNN) [40], Recurrent Neural Networks (RNN) [41], Generative Adversarial Networks (GAN) [42], etc.

Although deep learning approaches have several advantages, they also pose significant challenges in HSI applications. First of all, to achieve better classification result, often deep learning techniques demand large volumes of training samples. Moreover, a large number of hyper-parameters (like the kernel sizes, learning rate, etc.) are involved in training complex deep learning networks mainly designed for feature extraction and classification. Hence, the process becomes computationally expensive.

The disadvantages of combining SSA with structured approaches to incorporating spatial information may be overcome by using more flexible ways to spatially partition the dataset. In line with this consideration, in this work, a superpixel-based SSA (SP-SSA) algorithm was proposed as a means to increase the classifier performance. Instead of performing direct reconstruction, an object-specific reconstruction is performed to accurately preserve the local contextual information. Superpixel segmentation is first applied on the input HSI to generate a segmented HSI where each sub-region carries similar characteristic features, and its shape and size is adjusted according to the local image structure information. Next, 2D-SSA is individually applied on each segmented region to produce the reconstructed HSI. Lastly, the final classification map is generated by using the popular SVM classifier. The major novel contributions of this work are highlighted in the following list:

1.  Direct reconstruction is usually performed in standard 2D-SSA algorithms, where the full image is reconstructed. In HSIs, however, object-specific reconstruction is always better than direct reconstruction, as, in this way, local contextual information can be captured accurately. In this work, a novel SP-SSA approach is proposed that performs object-specific reconstruction.
2.  Superpixel segmentation and 2D-SSA are combined together for the first time for accurate spatial-spectral feature extraction. Using SP-SSA, each superpixel, i.e., object-specific spatial region is reconstructed.
3.  Superior classifier performance is achieved with the proposed method in comparison to other state-of-the-art methods, even with a comparatively small number of training samples.

The remainder of this paper is organized as follows. A detailed description of the proposed method is presented in Section 2. The experimental setup, results, and analysis are described in Section 3. Finally, some conclusions and future work are discussed in Section 4.

## 2. The Proposed Methodology

The proposed SP-SSA method includes three stages as described in Figure 1. In stage 1, superpixel segmentation is applied on the input HSI to obtain the segmented HSI. In stage 2, each segmented region is reconstructed using 2D-SSA to obtain the reconstructed HSI. In the final stage, an SVM classifier is applied on the reconstructed HSI to build the final classification map. A detailed description of each of these stages is presented in the subsections below.

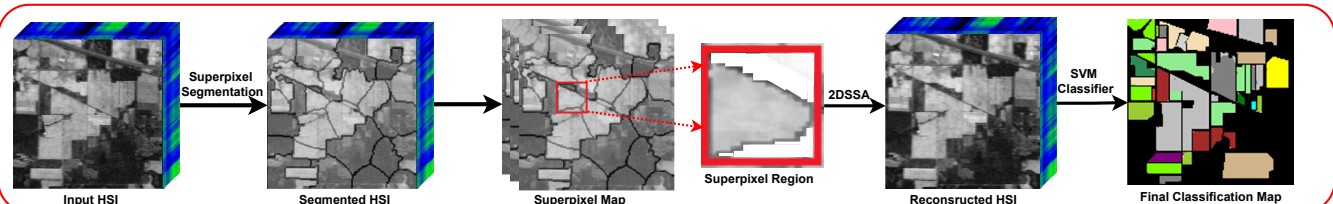

**Figure 1.** Flowchart of the proposed method.

### 2.1. Superpixel Segmentation

Superpixel segmentation approaches have gained popularity in recent years as these approaches have several benefits. Using superpixels, the computational complexity can be drastically reduced by computing features on more meaningful regions rather than acting on each individual pixel in HSI [43]. Simple Linear Iterative Clustering (SLIC) [44] is one of the most popular gradient-ascent-based superpixel segmentation approaches, where an initially defined tentative set of cluster points are iteratively refined using a gradient-ascent method until some convergence criteria are met. This algorithm has lower computational complexity as it applies the *k*-means method locally. The algorithm includes four key steps that can be summarized as follows.

The first step is cluster center initialization. Let the input HSI be denoted as $H^b \equiv \{h_1^b, h_2^b, \ldots, h_N^b\}$ with $N$ pixels, where $\{h_i^b\}$ represents the value at the $i$th pixel for the $b$th spectral band and $i = 1, 2, \ldots N$; $b = 1, 2, \ldots B$. $B$ is the total number of spectral bands. Each pixel can be labeled as $A_i = [h_i, r_i, u_i]$, where $h_i^T = [h_1, h_2, \ldots, h_B]^T$ is the spectral vector and $[r_i, u_i]^T$ is the position vector. The $K$ number of initial cluster centers $\mathcal{C}_j = [h_j, r_j, u_j]^T$ are sampled on a regular $Q \times Q$ ($Q = \sqrt{\frac{N}{K}}$) grid and are, thus, equally spaced apart [45].

The next step is the cluster assignment step, where each pixel is assigned to the nearby cluster center based on the computed distance measure $D$. Distance is computed within a $2Q \times 2Q$ window around the cluster center. The distance between the cluster center $\mathcal{C}_j$ and pixel $A_i$ is calculated as follows (Equation (1)):

$$D = D_{spectral} + \frac{w}{Q} D_{spatial} \tag{1}$$

where $w$ is the weighting factor between spectral and spatial features. The spectral and spatial distance between pixel $i$ and $j$ are represented as in Equations (2) and (3) below.

$$D_{spectral} = \sqrt{\sum_{b=1}^{B} \left( h_i^b - h_j^b \right)^2} \tag{2}$$

where $D_{spectral}$ is the measure of homogeneity within the superpixels.

$$D_{spatial} = \sqrt{(r_i - r_j)^2 + (u_i - u_j)^2} \tag{3}$$

where $(r_i, u_i)$ denotes the location of pixel $i$ in the superpixels. The spatial distance $D_{spatial}$ ensures regularity and compactness in the generated superpixels.

In the third step, the cluster centers are updated with the mean value of all pixels belonging to the same cluster. The second and third steps are iteratively repeated until convergence is achieved.

In the final step, post-processing is performed to enforce connectivity by reassigning disjoint pixels to nearby superpixels.

### 2.2. 2D-SSA

SSA is capable of decomposing a series into multiple independent components or subseries, where each extracted eigenvalue represents an individual component of the

original series. The SSA can be applied to the respective spectral bands of the hypercube, thereby, decomposing the 2-D scene, and then reconstructing it using the respective main components while removing the noise contribution. As a data cube is decomposed in this way, the local structure and main spatial trends are typically found in the first component. Hence, when all images within the hyperspectral cube are decomposed and only the first components are selected to individually reconstruct each of them, a resulting cube with minimum noise is generated. The SSA can be implemented using the following four steps:

### 2.2.1. Embedding

Imagine a HSI dataset $H$, with a size of $N_x \times N_y \times B$, where $N_x$, $N_y$ indicates the band image size and $B$ represents the total number of available bands. Each band image $H^b$ ($b \in B$) can be expressed as follows:

$$H^b = \begin{bmatrix} H^b_{1,1} & H^b_{1,2} & \cdots & H^b_{1,N_y} \\ H^b_{2,1} & H^b_{2,2} & \cdots & H^b_{2,N_y} \\ \vdots & \vdots & \ddots & \vdots \\ H^b_{N_x,1} & H^b_{N_x,2} & \cdots & H^b_{N_x,N_y} \end{bmatrix}_{N_x \times N_y} \tag{4}$$

Next, a 2D window $Q^b$ is defined, whose dimensions are $M_x \times M_y$.

$$Q^b = \begin{bmatrix} H^b_{i,j} & H^b_{i,j+1} & \cdots & H^b_{i,j+M_y-1} \\ H^b_{i+1,j} & H^b_{i+1,j+1} & \cdots & H^b_{i+1,j+M_y-1} \\ \vdots & \vdots & \ddots & \vdots \\ H^b_{i+M_x-1,j} & H^b_{i+M_x-1,j+1} & \cdots & H^b_{i+M_x-1,j+M_y-1} \end{bmatrix}_{M_x \times M_y} \tag{5}$$

where $1 \leq M_x \leq N_x$, $1 \leq M_y \leq N_y$, and $1 < M_x M_y < N_x N_y$. Each pixel is spatially positioned by $(i, j)$ within the image $H^b$. The pixels in a window $Q^b$ can be rearranged into a column vector $C^b_{i,j} \in \mathbb{R}^{M_x M_y}$ according to the reference position $(i, j)$ as follows:

$$C^b_{i,j} = [H^b_{i,j}, H^b_{i,j+1}, \ldots, H^b_{i,j+M_y-1}, H^b_{i+1,j}, H^b_{i+1,j+1}, \ldots, H^b_{i+M_x-1,j+M_y-1}]^T \tag{6}$$

To scan the whole image $H^b$, this 2-D window is slid across it from top left to bottom right until it has visited every position on the entire image (see also Figure 2 for a graphical explanation).

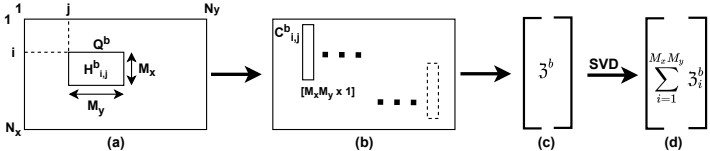

**Figure 2.** Moving window across the image $H^b$ to create the trajectory matrix $\mathfrak{Z}^b$.

As a result, the trajectory matrix $\mathfrak{Z}^b$ of all feasible 2-D windows of image $H^b$ of size $M_x M_y \times (N_x - M_x + 1)(N_y - M_y + 1)$ can be obtained as follows:

$$\mathfrak{Z}^b = \left[ (C^b_{1,1})^T, (C^b_{1,2})^T, \ldots, (C^b_{1,N_y-M_y+1})^T, (C^b_{2,1})^T, \ldots, (C^b_{N_x-M_x+1,N_y-M_y+1})^T \right]_{M_x M_y \times (N_x-M_x+1)(N_y-M_y+1)} \tag{7}$$

Note that the trajectory matrix $\mathfrak{Z}^b$ has a structure of Hankel–block–Hankel (HbH). $\mathfrak{Z}^b$ can be expressed as follows:

$$\mathfrak{Z}^b = \begin{bmatrix} P_1^b & P_2^b & \cdots & P_{N_x-M_x+1}^b \\ P_2^b & P_3^b & \cdots & P_{N_x-M_x+2}^b \\ \vdots & \vdots & \ddots & \vdots \\ P_{M_x}^b & P_{M_x+1}^b & \cdots & P_{N_x}^b \end{bmatrix}_{M_x \times (N_x-M_x+1)} \tag{8}$$

Each of the submatrices $P_i^b$ corresponds to a Hankel structure as follows:

$$P_i^b = \begin{bmatrix} H_{i,1}^b & H_{i,2}^b & \cdots & H_{i,N_y-M_y+1}^b \\ H_{i,2}^b & H_{i,3}^b & \cdots & H_{i,N_y-M_y+1}^b \\ \vdots & \vdots & \ddots & \vdots \\ H_{i,M_y}^b & H_{i,M_y+1}^b & \cdots & H_{i,N_y}^b \end{bmatrix}_{M_y \times (N_y-M_y+1)} \tag{9}$$

### 2.2.2. Singular Value Decomposition (SVD)

After obtaining the trajectory matrix $\mathfrak{Z}^b$, SVD is applied to determine the eigenvalues $\left(\lambda_1 \geq \lambda_2 \geq \cdots \geq \lambda_{M_x M_y}\right)$, and the corresponding eigenvectors $\left(U_1, U_2, \cdots, U_{M_x M_y}\right)$ of $\left(\mathfrak{Z}^b\left(\mathfrak{Z}^b\right)^T\right)$. It is possible to rewrite $\mathfrak{Z}^b$ as follows:

$$\mathfrak{Z}^b = \mathfrak{Z}_1^b + \mathfrak{Z}_2^b + \cdots + \mathfrak{Z}_{M_x M_y}^b \tag{10}$$

where the $i$th elementary matrix is $\mathfrak{Z}_i^b = \sqrt{\lambda_i} U_i V_i^T$ and its Principal Components (PCs) are defined as $V_i = \frac{(\mathfrak{Z}^b)^T U_i}{\sqrt{\lambda_i}}$

### 2.2.3. Grouping

A subsequent operation is eigenvalue grouping, during which the total set of $M_x M_y$ individual components in (10) are divided into $m$ subsets, designated as $S = [S_1, S_2, \ldots, S_m]$. By selecting one or more elementary matrices $\mathfrak{Z}_i^b$ from each subset, it is possible to derive the main information contained in an image without being disturbed by high noise levels. As a result, the trajectory matrix $\mathfrak{Z}^b$ can be represented as follows:

$$\mathfrak{Z}^b = \mathfrak{Z}_{S_1}^b + \mathfrak{Z}_{S_2}^b + \cdots + \mathfrak{Z}_{S_m}^b \tag{11}$$

The reconstruction of a single band scene of HSI using various numbers of components $(\mathfrak{Z}_i^b)$ is compared in Figure 3. In general, the component with the highest eigenvalue is the most informative one, containing key features with the lowest noise contribution. With the inclusion of additional components, the reconstructed scene begins to resemble the actual scene. The reconstructed image obtained by grouping the 1st–5th components and 1st–10th components are very similar with marginal differences (Figure 3c,d). Hence, a small number of key components are sufficient to reconstruct the scene satisfactorily.

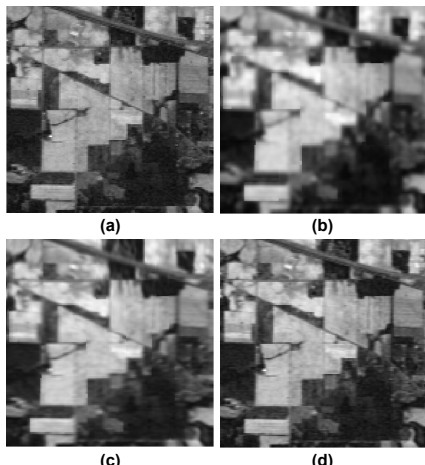

**Figure 3.** Implementation of 2D-SSA on a HSI scene (**a**) Original scene at 667 nm. (**b**) 1st component grouping. (**c**) 1–5th component grouping. (**d**) 1–10th component grouping, where $M_x = 5$, and $M_y = 5$.

### 2.2.4. Diagonal Averaging

$\mathfrak{Z}^b$, in this case, does not necessarily belong to the HbH matrix type. It is projected into a 2D-signal by applying the Hankelization process in two steps; first inside every block (9) and next block-to-block (8) by averaging the anti-diagonal elements in the matrix. Thus, it is possible to obtain a reconstructed image that contains the distinctive spatial features based on the local contextual information present in a 2D window defined by the user.

### 2.3. Novelty of the Proposed SP-SSA Method

The proposed approach integrates SSA and superpixel segmentation for the first time to extract improved the spatio-spectral features from HSI. Reconstruction of object-specific spatial sections, rather than the entire image, may be more effective. Hence, in the proposed work, 2D-SSA is applied individually to each superpixel segmented region to extract the local contextual information accurately. The pseudo-code for the proposed SP-SSA algorithm is outlined in Algorithm 1.

---

**Algorithm 1:** Proposed SP-SSA algorithm for HSI classification.

---

**Input:** HS image, $H \in \mathbb{R}^{n \times b}$
  Ground Truth $GT$
  number of superpixels: K
  Embedding Window Size: $M_x \times M_y$
  Eigen Value Grouping: $EV$
**Output:** Classification Map *clsmap* generated by SVM.
  1: **for** $b = 1$ to $B$ **do**
  2:   Perform SLIC superpixel segmentation to obtain segmentation map $L$ from $h^b$
      containing $K$ superpixel segments
      $L = $ SLIC $(h^b, K)$    (as outlined in Section 2.1)
  3:   **for** $k = 1$ to $K$ **do**
  4:     $h_k^{b'} = reconstruct2DSSA(L, M_x, M_y, EV)$    (as outlined in Section 2.2)
  5:   **end for**
  6: **end for**
  7: Obtained Reconstructed HSI $H' \in \mathbb{R}^{n \times b}$
  8: $clsmap = $ SVM $(H', GT)$

---

For each superpixel, the reconstruction (*reconstruct2DSSA* (Algorithm 1)) is applied to the rectangular Region of Interest (ROI) surrounding the superpixel (Figure 4). The ROI is created based on the location information of the pixels available in that particular segment. Only the reconstructed pixels specific to those pixels in the selected superpixel are

stored as spatial features, while the remaining reconstructed pixels in the ROI are discarded as they do not belong to the superpixel under test. The same procedure is applied to all other superpixels, and the HS image is reconstructed using the proposed SP-SSA approach. This procedure collects local object-specific superpixel-based spatial features for each band in the image.

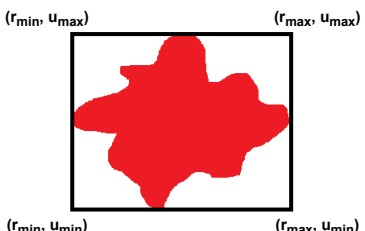

**Figure 4.** Possible Region of Interest (ROI) around the superpixel segment. $(r, u)$ denotes the location of pixel $i$ in superpixels. $r_{min}$, $r_{max}$ represents the minimum and maximum row index, and $u_{min}$, $u_{max}$ are the corresponding *min* and *max* column indices.

### 2.4. Classification

The selection of an appropriate classifier is critical in assessing the performance of the above-mentioned features, especially in hyperspectral images with a limited number of training samples. SVM is the most widely used supervised statistical learning framework among pixel-wise classifiers. With the help of a kernel function, data can be mapped to a higher-dimensional space via a nonlinear transformation, aiming to determine the best hyperplane for separating samples belonging to different classes. The performance of SVM in HSI classification is outstanding despite the variation of the data dimensions [46,47]. Hence, in this work, the SVM classifier is utilized to evaluate the performance of the reconstructed features.

## 3. Results and Discussion

This section reports the outcome of testing the proposed approach on some of the most popular benchmark datasets and compares it with other, state-of-art classification approaches.

### 3.1. Dataset Description

In this subsection, the datasets used for testing the proposed approach are presented and described.

#### 3.1.1. Indian Pines

The first dataset, named "Indian Pines" (IP), was collected over Northwestern Indiana, USA, with the airborne AVIRIS sensor; it includes a total of 220 bands covering wavelengths from 0.4 to 2.5 μm. About 70% of the imaged area is agricultural land, while the remaining portions are forests. Due to the comparatively low spatial resolution (20 m/pixel) of the sensor, this dataset is challenging as it contains highly mixed pixels. The number of samples obtained per class is also unbalanced, which further complicates classification. The size of the scene is 145 × 145 pixels, and its Ground Truth (GT) data defines 16 different classes. The pseudo-color image, the GT map, and the class names for the dataset are all included in Figure 5.

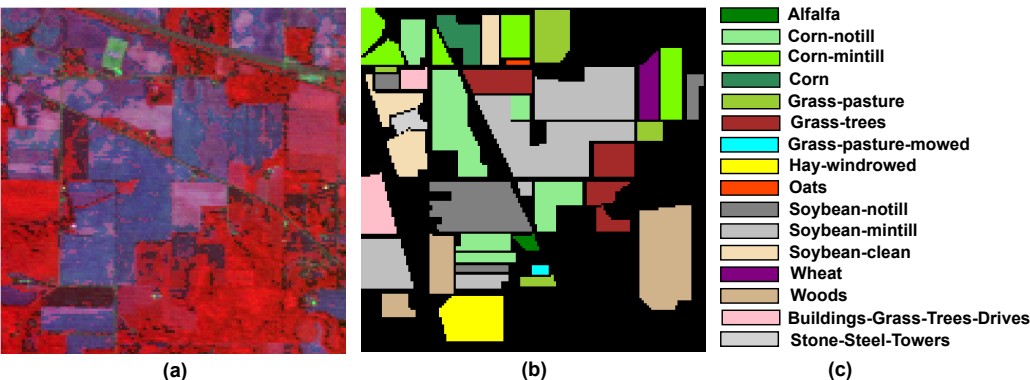

**Figure 5.** (**a**) False Color Composite Image, (**b**) Ground Truth Image and (**c**) Class names for the Indian Pines Dataset.

### 3.1.2. Pavia University

The ROSIS sensor was instrumental to the collection of this dataset over the University of Pavia, Italy. The dataset is called "Pavia University" (PU). It has a spatial resolution of 1.3 m and originally comprises 115 spectral bands covering wavelength ranges from 0.43 to 0.86 μm. In the final analysis, 103 bands are used after the elimination of noisy channels. The image has a size of 610 × 340 pixels, and it has nine challenging classes with nearly similar spectral reflectances. Detailed information about the false-color image, Ground Truth, and class names is displayed in Figure 6.

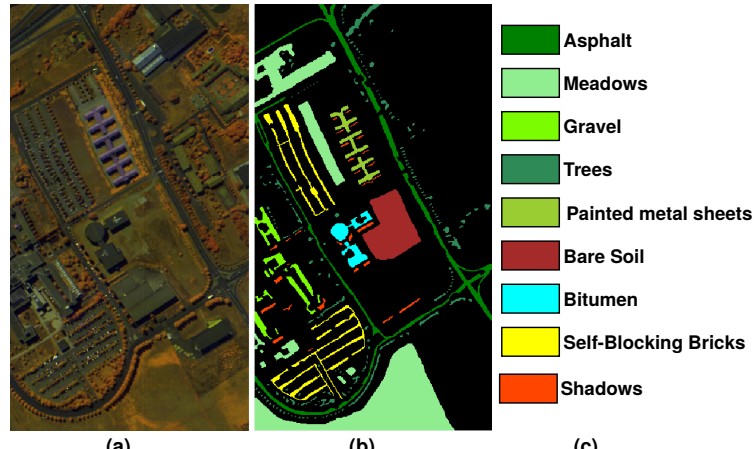

**Figure 6.** (**a**) False Color Composite Image, (**b**) Ground Truth Image and (**c**) Class names for the Pavia University Dataset.

### 3.1.3. Salinas Dataset

The "Salinas" (SAL) dataset was captured over the Salinas Valley, California, USA, using the AVIRIS Sensor. The sensor has 224 channels with spectral range varying from 0.43 μm to 2.5 μm. This scene has a size of 512 ×217 pixels and spatial resolution of 3.7 m per pixel. The number of bands reduces to 204 after discarding 20 water absorption bands: [108–112], [154–167], 224. The scene is mainly an agricultural area, with 16 classes in its Ground Truth. A false color representation, the Ground Truth, and the class names for the Salinas dataset are shown in Figure 7.

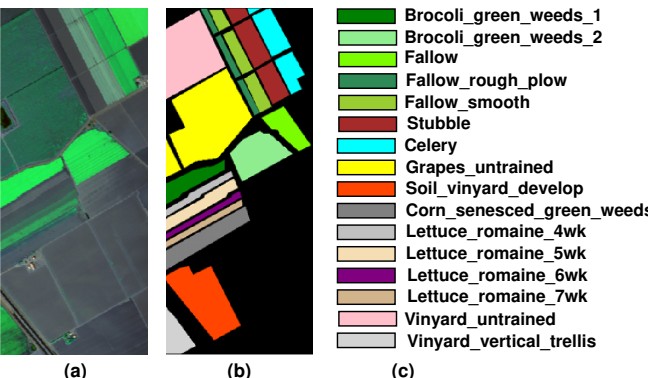

**Figure 7.** (**a**) False Color Composite Image, (**b**) Ground Truth Image, and (**c**) Class names for the Salinas Dataset.

### 3.1.4. Houston 2018

The 2018 IEEE GRSS Data Fusion Contest (DFC) triggered public dissemination of this rich dataset, which was included in our tests to increase their statistical significance. The image of the Houston campus and its surrounding area was captured by the IRTES CASI-1500 sensor at a GSD of 1 m over Houston, Texas, USA. It has 601 × 2384 pixels and 50 spectral bands with wavelengths ranging from 380 to 1050 nm sampled at 10 nm intervals. The scene contains 20 urban landcover classes. The false-color composite image, ground truth image, and class names for the Houston 2018 dataset are provided in Figure 8.

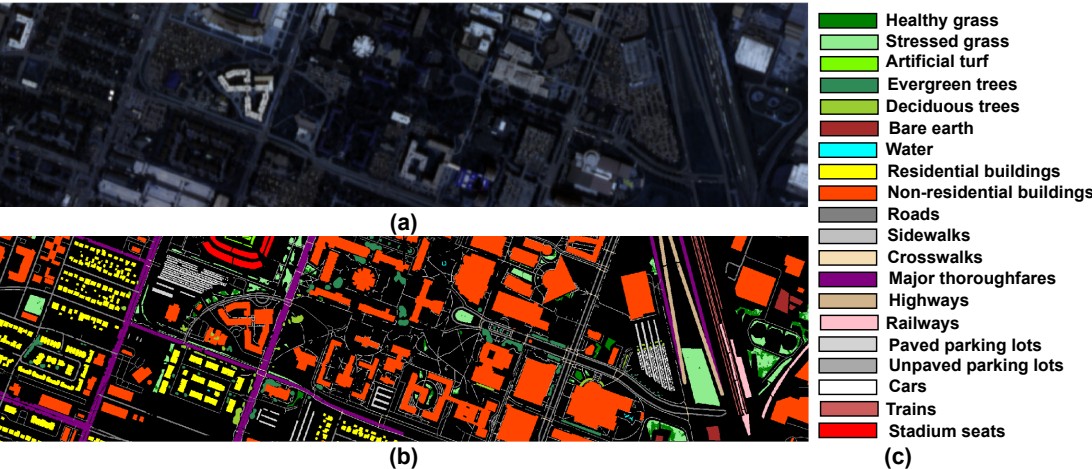

**Figure 8.** (**a**) False-Color Composite Image, (**b**) Ground Truth Image, and (**c**) Class names for the Houston-2018 Dataset.

### 3.2. Experimental Setup

Our proposed approach was evaluated by comparing its performance with eight state-of-the-art approaches for HSI feature extraction (Algorithm 2, see Section 3.4.5). These include SVM [7], Edge Preserving Filter (EPF) [26], superpixel-based classification via multiple kernels (SCMK) [34], region-based relaxed multiple kernel (R2MK) [35], adjacent superpixel-based multiscale generalized spatial-spectral kernel (ASMGSSK) [36], Multiscale superpixel-based PCA (MsuperPCA) [37], 2D Singular Spectrum Analysis (2D-SSA) [29], and 2D Multiscale Singular Spectrum Analysis (2D-MSSA) [31]. A common way to measure the efficiency of feature extraction is through the accuracy of the classifier scored by the experiments. As a result, the classification setup must be appropriate with the current state-of-the-art. In light of this, SVMs have demonstrated themselves to be robust and efficient in multi-class classification applications.

The LIBSVM toolbox [48] is used to implement SVM as the default classifier for all of the involved methods. A Gaussian RBF kernel is utilized for SVM implementation,

and a grid search is applied to tune both key parameters of RBF-SVM; the penalty *c* and the gamma $\gamma$. The SVM parameters are kept constant across all competitive experiments for a fair comparison. To avoid systematic errors and reduce random discrepancies, all experiments were independently carried out ten times each with different training and testing subsets, with no overlap between each training and the corresponding testing subset. This was intended to ensure good statistical significance for our experiments.

Stratified sampling was used to randomly obtain the training and testing subsets. For training, 3%, 2%, 1%, and 0.2% samples per class were selected for the IP, PU, SAL, and Houston 2018 datasets, respectively. Additionally, four objective quality indices are utilized to evaluate image classification results: namely the OA, the average accuracy (AA), the kappa coefficient, and class-by-class accuracy. All experiments were conducted using MATLAB R2018b software, installed on a personal computer with an Intel core i5-6200 CPU clocked at 2.30 GHz, and 16 GB RAM.

### 3.3. Parameter Sensitivity Analysis

Table 1 displays the best parameter settings for the competing algorithms, found by experimentation. For the proposed SP-SSA algorithm, the size of the 2-D embedding window was set to $5 \times 5$ pixels for the IP and Salinas dataset; whereas, for the PU and Houston 2018 dataset, the window size was set at $3 \times 3$ pixels. For the IP and SAL datasets, superpixels were set at 100. However, the amount of superpixels in the PU and Houston 2018 datasets were set to 150 and 500, respectively. The effect of window size variation for different number of superpixels on the classification performance for the experimental datasets is provided in Figure 9.

As each superpixel is reconstructed individually, smaller window sizes are preferred since they lead to better image reconstructions. Using a large window may smooth the results too much and result in mixing errors. A 2D-SSA algorithm was presented in [29] for feature extraction in HSI, where various window sizes, such as $5 \times 5$, $10 \times 10$, $20 \times 20$, $40 \times 40$, and $60 \times 60$, were examined. The IP and SAL datasets produced the best classification accuracy when the window size was set at $10 \times 10$. When analyzing the PU and Houston dataset, the window sizes of $5 \times 5$ showed the best classification results. Since the optimal window size may vary depending on the dataset, ref. [31] adopts a multiscale strategy to improve the generalization ability.

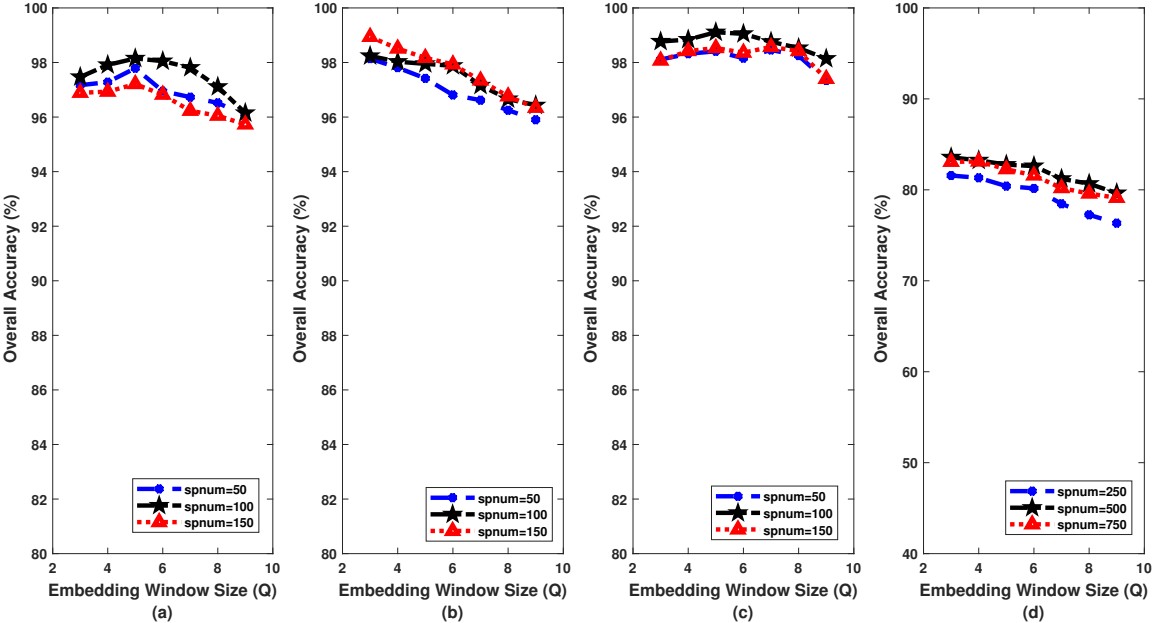

**Figure 9.** Effect of window size variation for different number of superpixels on the classification performance for the (**a**) Indian Pines (**b**) Pavia University, and (**c**) Salinas, and (**d**) Houston 2018 datasets.

**Table 1.** Parameter settings for different algorithms in the Indian Pines, Pavia University, Salinas, and Houston 2018 Datasets.

| Method | Indian Pines, Salinas | Pavia University | Houston 2018 |
|---|---|---|---|
| SVM | NA | NA | NA |
| EPF | $\delta_s = 3$, $\delta_r = 0.2$, r = 3, and $\epsilon = 0.01$ | $\delta_s = 3$, $\delta_r = 0.2$, r = 3, and $\epsilon = 0.01$ | $\delta_s = 5$, $\delta_r = 0.1$, r = 3, and $\epsilon = 0.02$ |
| SCMK | $\sigma = 2^{-6}$; spnum = 600; $\mu_1 = 0.2$, $\mu_2 = 0.2$ | $\sigma = 2^{-4}$; spnum = 900; $\mu_1 = 0.2$, $\mu_2 = 0.4$ | $\sigma = 2^{-5}$; spnum = 1600; $\mu_1 = 0.1$, $\mu_2 = 0.3$ |
| R2MK | $\sigma = 2^{-6}$; spnum = [20,50,100,200,400,800]; $\mu = 0.2$ | $\sigma = 2^{-5}$; spnum = [50,100,200,400,800,1600]; $\mu = 0.3$ | $\sigma = 2^{-4}$; spnum = [50,100,200,400,800,1600,3200]; $\mu = 0.1$ |
| ASMGSSK | $r_0 = 0.1$; $\sigma = 2^{-7}$; spnum = [100,200,400,800,1600,3200] | $r_0 = 0.1$; $\sigma = 2^{-5}$; spnum = [200,400,800,1600,3200,6400] | $r_0=0.2$; $\sigma = 2^{-4}$; spnum = [200,400,800,1600,3200,6400] |
| MsuperPCA | Fundamental spnum = 100; scale no = 4 | Fundamental spnum = 20; scale no = 6 | Fundamental spnum = 100; scale no = 8 |
| 2DSSA | Window Size: $10 \times 10$; EVG = 1st | Window Size: $5 \times 5$; EVG = 1 − 2nd | Window Size: $5 \times 5$; EVG = 1 − 2nd |
| 2DMSSA | Window Size: $5 \times 5, 10 \times 10, 20 \times 20, 40 \times 40, 60 \times 60$ | Window Size: $5 \times 5, 10 \times 10, 20 \times 20, 40 \times 40, 60 \times 60$ | Window Size: $5 \times 5, 10 \times 10, 20 \times 20, 40 \times 40, 60 \times 60$ |
| SP-SSA | spnum: 100; Window Size: $5 \times 5$; EVG = 1 | spnum: 150; Window Size: $3 \times 3$; EVG = 1 − 2nd | spnum: 500; Window Size: $3 \times 3$; EVG = 1 − 2nd |

### 3.4. Experimental Result and Analysis

In this section, the four HSI data sets outlined in Section 3.1 are utilized, and several experiments are performed to examine the efficacy of the proposed SP-SSA method. Figure 10 compares the classification results obtained with varying numbers of training samples on four datasets. It can be noted that better classification performance is evident when larger numbers of labeled samples are utilized for training; after passing the percentages used in this work; however, the accuracy level mostly plateaus, and no further significant improvement is observed. Our proposed approach attains the best classification accuracy in almost all cases, regardless of the number of samples, proving its robustness. Classification results from all four data sets are provided in Tables 2–5 and quantitatively support the dominance of the proposed method. Individual classification maps generated by the proposed SP-SSA method and all the compared approaches are displayed in Figures 11–14.

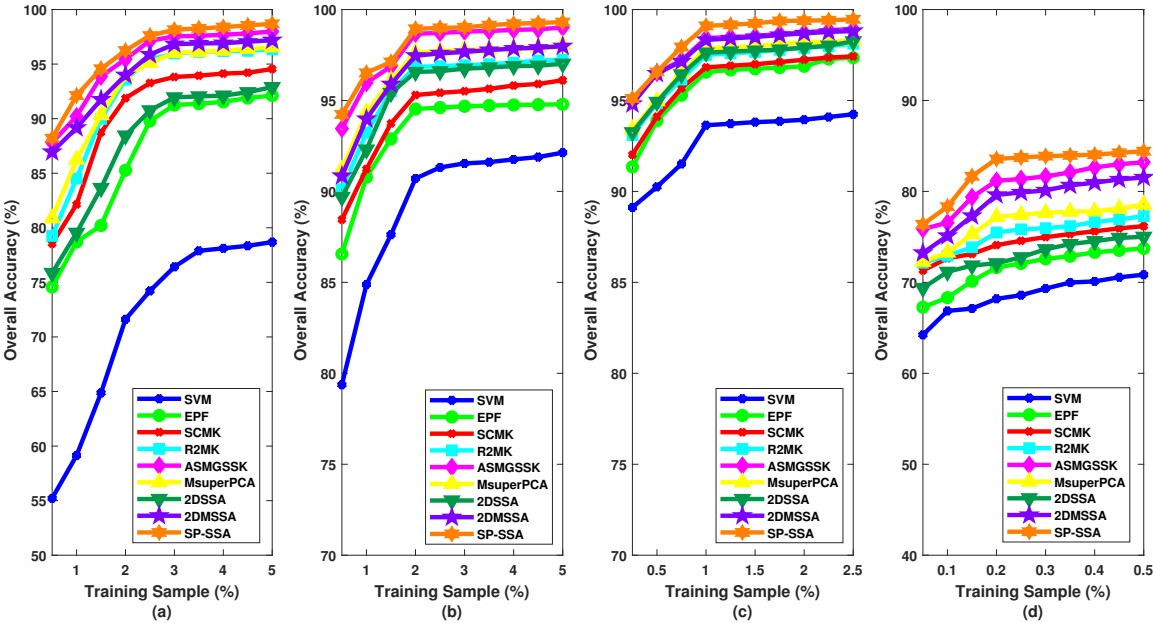

**Figure 10.** Effect of training sample variation on the classification performance for the (**a**) Indian Pines (**b**) Pavia University, (**c**) Salinas, and (**d**) Houston 2018 datasets.

#### 3.4.1. Results from the Indian Pines Dataset

Based on the results shown in Table 2, the proposed method achieves the best values across three metrics, and its accuracy exceeded 89% on almost all classes. In the tables, the best results in each row are highlighted in bold font. When comparing SP-SSA with raw HSI data, the OA improved substantially from 76.42% to 98.15%. In addition, comparisons between SVM and other methods indicated that the incorporation of spatial features can enhance the classification performance compared to considering spectral features alone.

Superpixel-based methods, such as SCMK, R2MK, ASMGSSK, and MsuperPCA techniques, yield higher classification accuracy as compared to non-superpixel based techniques (EPF, 2D-SSA, and 2D-MSSA); by grouping spectrally identical regions, superpixels offer a powerful way to exploiting spatial/contextual information. It can also be noted that methods considering multi-scale windows (ASMGSSK, MsuperPCA, and 2D-MSSA) perform better with respect to fixed-window methods. Due to the different window sizes, unique local spatial features can be exploited, which allows better covering of different sizes of land cover classes and different scales of spatial features. On the downside, the use of multiscale approaches involves heavier processing burdens. As the proposed method reconstructs each superpixel individually, better classification results are obtained.

**Table 2.** Classification results for the Indian Pines Dataset with 3% training for SVM, EPF, SCMK, R2MK, ASMGSSK, MsuperPCA, 2D-SSA, 2D-MSSA, and SP-SSA algorithms.

| Class | Samples | SVM | EPF | SCMK | R2MK | ASMGSSK | MsuperPCA | 2DSSA | 2DMSSA | SP-SSA |
|---|---|---|---|---|---|---|---|---|---|---|
| Alfalfa | 46 | 12.65 | 22.22 | 62.79 | 92.86 | **95** | 87.8 | 54.55 | 87.8 | 92.11 |
| Corn-notill | 1428 | 75.18 | 92.85 | 92.55 | 94.06 | 94.85 | 97.09 | 91.81 | 95.25 | **98.25** |
| Corn-mintill | 830 | 82.46 | 83.44 | 90 | 93.32 | 97.37 | 90.11 | 88.96 | 96.65 | **97.99** |
| Corn | 237 | 47.86 | 79.39 | 85.59 | 95.41 | 96.14 | **99.53** | 80.89 | 89.2 | 96.98 |
| Grass-pasture | 483 | 69.25 | 68.04 | 89.67 | 97.3 | **99.05** | 95.81 | 90.61 | 92.41 | 97.04 |
| Grass-trees | 730 | 80.19 | 96.72 | **99.56** | 98.36 | 99.21 | 98.61 | 93.23 | 99.54 | 99.51 |
| Grass-pasture-mowed | 28 | 88.89 | 88.46 | 69.23 | 80 | **100** | 96 | 92.59 | 92 | **100** |
| Hay-windrowed | 478 | 93.46 | **100** | 98 | 98.18 | 98.32 | 95.07 | 98.46 | 99.54 | 99 |
| Oats | 20 | 36.12 | 47.37 | **100** | 52.63 | 47.06 | 94.12 | 42.11 | 83.33 | **100** |
| Soybean-notill | 972 | 76.74 | 82.96 | 88.29 | 90.16 | 96.57 | 95.27 | 87.12 | **96.91** | 93.63 |
| Soybean-mintill | 2455 | 79.51 | 97.03 | 95.54 | 97.48 | 98.5 | 96.66 | 93.1 | 97.19 | **99.03** |
| Soybean-clean | 593 | 50 | 86.99 | 93.18 | 95.79 | 97.29 | 95.27 | 90.96 | 96.07 | **98.6** |
| Wheat | 205 | 99.01 | 96.94 | 92.23 | **100** | 97.21 | 98.9 | 99.48 | 98.91 | **100** |
| Woods | 1265 | 88.74 | 99.26 | 98.91 | 99.48 | 99.27 | 99.02 | 96.42 | **99.82** | 99.62 |
| Buildings-Grass-Trees-Drives | 386 | 65.18 | 87.87 | **99.45** | 94.37 | 96.73 | 88.95 | 90.16 | 94.24 | 98.46 |
| Stone-Steel-Towers | 93 | 41.32 | 97.75 | 72.73 | 92.94 | 97.5 | 93.9 | 97.75 | **98.81** | 89.74 |
| **OA::** | | 76.42 | 91.25 | 93.82 | 95.98 | 97.54 | 96.04 | 91.95 | 96.83 | **98.15** |
| **AA::** | | 67.91 | 82.96 | 89.23 | 92.02 | 94.38 | 95.13 | 86.76 | 94.86 | **97.5** |
| **K::** | | 73.05 | 89.98 | 92.95 | 95.32 | 97.2 | 95.49 | 90.82 | 96.39 | **97.89** |

Figure 11 displays the classification maps produced by various approaches for the Indian Pines dataset. For the SVM approach, the classification map appears very noisy if spatial features are not considered. Through the use of neighborhood spatial information, the EPF and 2D-SSA techniques can suppress spot-wise misclassification to a large extent, but these methods do not preserve the detailed structures of the HSI well enough.

However, by adopting superpixel-based approaches, the generated classification map becomes much smoother, and more accurate estimates are obtained in the detailed region. With the utilization of multi-scale approaches (like ASMGSSK, MSuperPCA, and 2D-MSSA), the amount of misclassification is further reduced. Still, even with multi-scale approaches, landcover boundaries are frequently misplaced. As can be observed from Figure 11, the proposed approach effectively solved the above-mentioned problems due to its considerate utilization of spectral and spatial features.

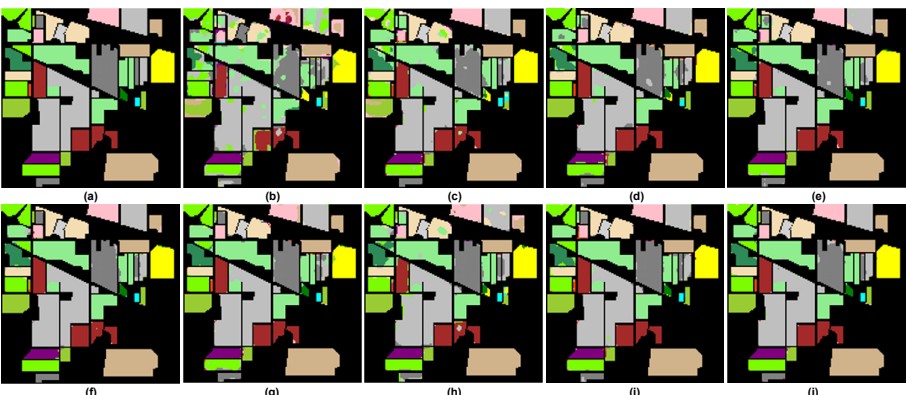

**Figure 11.** (**a**) Ground Truth Image, Classification Maps of (**b**) SVM (**c**) EPF (**d**) SCMK (**e**) R2MK (**f**) ASMGSSK (**g**) MsuperPCA (**h**) 2D-SSA (**i**) 2D-MSSA (**j**) SP-SSA for Indian Pines dataset.

3.4.2. Results from the Pavia University Dataset

Quantitative results are presented in Table 3. The proposed SP-SSA method still achieved higher classification accuracy and ranked first among all the compared methods, closely followed by the ASMGSSK algorithm. Also, in comparison to EPF, SCMK, R2MK, 2D-SSA, MSuperPCA, and 2D-MSSA techniques, the average improvement of the proposed approach is over 4.41%, 3.64%, 2.09%, 2.37%, 1.3%, and 1.48%, respectively. For comparison, the top results in the tables are boldfaced. In Figure 12, different classification maps are shown, based on various testing methods applied to the PU dataset.

According to Figure 12, the classification map for SVM still continues to remain noisy. Both EPF and 2D-SSA can generate a relatively smooth result; however, some significant regions remain undetected (e.g., the detailed areas). The superpixel-based methods (SCMK, R2MK, ASMGSSK, and MSuperPCA) and SSA-based approach (2D-SSA and 2D-MSSA) offer significantly improved performance, but the proposed 2D-SSA method remains the most promising approach as it outperforms all the compared algorithms.

**Table 3.** Classification results for the Pavia University Dataset with 2% training for SVM, EPF, SCMK, R2MK, ASMGSSK, MsuperPCA, 2D-SSA, 2D-MSSA, and SP-SSA algorithms.

| Class | Samples | SVM | EPF | SCMK | R2MK | ASMGSSK | MsuperPCA | 2DSSA | 2DMSSA | SP-SSA |
|---|---|---|---|---|---|---|---|---|---|---|
| Asphalt | 6631 | 90.79 | 96.33 | 94.01 | 95.43 | **99.23** | 97.19 | 97.12 | 97.61 | 98.94 |
| Meadows | 18,649 | 99.54 | 98.06 | 99.14 | 99.78 | 99.76 | 99.83 | 99.52 | 99.45 | **99.85** |
| Gravel | 2099 | 53.12 | 80.86 | 83.69 | 90.81 | 91.85 | 89.73 | 89.35 | 90.53 | **94.17** |
| Trees | 3064 | 81.9 | 83.81 | 87.34 | 93.8 | 98.22 | 95.53 | 96.24 | 95.76 | **99** |
| Painted metal sheets | 1345 | 93.04 | **100** | 99.17 | **100** | 99.17 | **100** | 99.47 | 99.46 | **100** |
| Bare Soil | 5029 | 89.64 | 94.15 | 96.46 | 96.81 | 99.6 | 98.01 | 95.76 | 98.52 | **99.74** |
| Bitumen | 1330 | 55.48 | 89.74 | 86.86 | 91.71 | **99.42** | 94.54 | 85.97 | 97.21 | 98.85 |
| Self-Blocking Bricks | 3682 | 87.92 | 91.47 | 90.97 | 90.97 | 94.69 | 93.05 | 89.11 | 92.64 | **95.62** |
| Shadows | 947 | 90.75 | 90.19 | 97.12 | 98.17 | 98.59 | **98.89** | 95.69 | 88.67 | 98.76 |
| **OA::** | | 90.71 | 94.53 | 95.3 | 96.85 | 98.67 | 97.64 | 96.57 | 97.46 | **98.94** |
| **AA::** | | 82.47 | 91.62 | 92.75 | 95.28 | 97.84 | 96.31 | 94.25 | 95.54 | **98.33** |
| **K::** | | 87.57 | 92.73 | 93.76 | 95.82 | 98.23 | 96.87 | 95.44 | 96.63 | **98.6** |

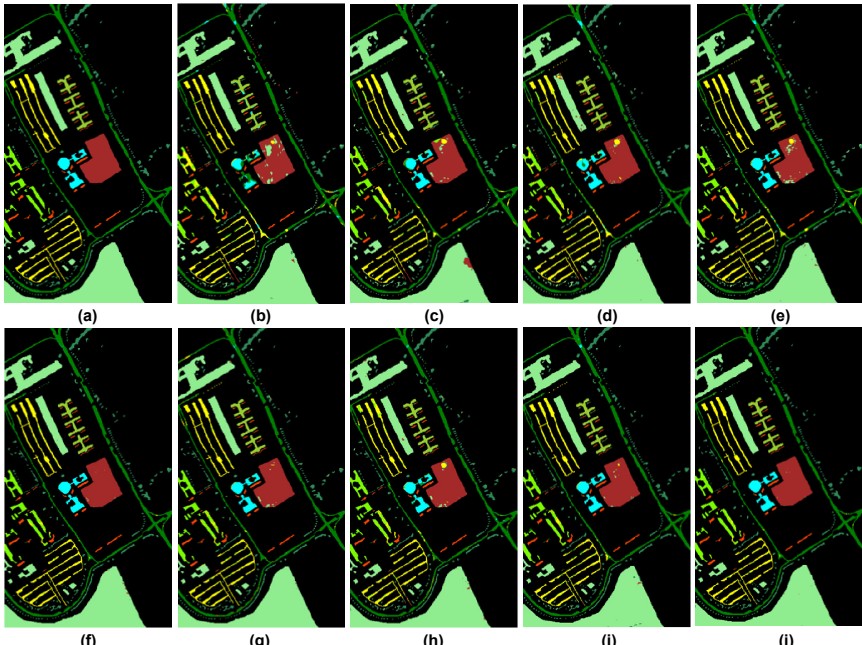

**Figure 12.** (**a**) Ground Truth Image, Classification Maps of (**b**) SVM (**c**) EPF (**d**) SCMK (**e**) R2MK (**f**) ASMGSSK (**g**) MsuperPCA (**h**) 2D-SSA (**i**) 2D-MSSA (**j**) SP-SSA for the Indian Pines dataset.

### 3.4.3. Results from the Salinas Dataset

The visual classification maps and quantitative results obtained by various classifiers on the Salinas dataset are shown in Figure 13 and Table 4, respectively. In the table, the best results are shown in bold. Based on the visual quality as well as objective metrics, it can be observed that the proposed SP-SSA method outperformed other competing approaches. In addition, compared with the 2D-SSA method that globally reconstructs the image using fixed-size embedded windows, the SP-SSA method considers the local spatial information by reconstructing each superpixel individually, which helps in further reducing the disturbances and improving the class assignment.

**Table 4.** Classification Results for the Salinas Dataset with 1% training for SVM, EPF, SCMK, R2MK, ASMGSSK, MsuperPCA, 2D-SSA, 2D-MSSA, and SP-SSA algorithms

| Class | Samples | SVM | EPF | SCMK | R2MK | ASMGSSK | MsuperPCA | 2DSSA | 2DMSSA | SP-SSA |
|---|---|---|---|---|---|---|---|---|---|---|
| Brocoli-green-weeds-1 | 2009 | 99.74 | 98.43 | 99.44 | 99.85 | 99.36 | 99.39 | 98.83 | **100** | 99.64 |
| Brocoli-green-weeds-2 | 3726 | **100** | 99.92 | **100** | 98.62 | **100** | 99.82 | **100** | **100** | 99.96 |
| Fallow | 1976 | 93.34 | 99.31 | 99.9 | **100** | 99.71 | 99.89 | 99.64 | 99.64 | **100** |
| Fallow-rough-plow | 1394 | 96.37 | 97.28 | 98.52 | 98 | 98.26 | 98.56 | 97.38 | 97.54 | **98.67** |
| Fallow-smooth | 2678 | 91.19 | 99.06 | 97.77 | 99.42 | 98.72 | 99.75 | 98.42 | 99.25 | **99.63** |
| Stubble | 3959 | **100** | 99.92 | 98.54 | 99.92 | **100** | **100** | **100** | **100** | **100** |
| Celery | 3579 | 99.18 | 99.65 | 99.14 | 99.54 | 99.8 | 99.78 | **100** | 99.88 | 99.96 |
| Grapes-untrained | 11,271 | 91.38 | 92.49 | 94.6 | 95.2 | 96.77 | 95.76 | 95 | 96.31 | **97.91** |
| Soil-vinyard-develop | 6203 | 97.34 | 99.88 | 99.65 | 99.98 | 99.82 | 99.96 | 99.66 | 99.95 | **100** |
| Corn-senesced-green-weeds | 3278 | 93.96 | 97.37 | **98.96** | 97.42 | 98.47 | 98.24 | 96.73 | 98.3 | 98.82 |
| Lettuce-romaine-4wk | 1068 | 81.66 | 96.55 | 96.81 | 97.78 | 97.99 | 97.92 | 97.24 | 97.86 | **99.87** |
| Lettuce-romaine-5wk | 1927 | 98.03 | **100** | **100** | **100** | **100** | **100** | **100** | **100** | **100** |
| Lettuce-romaine-6wk | 916 | 98.85 | 97.01 | 97.41 | 99.55 | 99.69 | **99.76** | 99.74 | 99.69 | 99.69 |
| Lettuce-romaine-7wk | 1070 | 92.92 | 98.23 | 99.13 | 98.17 | 99.07 | 98.86 | **99.78** | 99.47 | 99.73 |
| Vinyard-untrained | 7268 | 82.69 | 90.41 | 91.32 | 92.28 | 95.89 | 92.98 | 94.21 | 95.58 | **97.7** |
| Vinyard-vertical-trellis | 1807 | 95.63 | 99.13 | 90.3 | 99.83 | 99.6 | 99.82 | 98.57 | 99.29 | **99.92** |
| **OA::** | | 93.64 | 96.56 | 96.82 | 97.5 | 98.43 | 97.89 | 97.63 | 98.33 | **99.1** |
| **AA::** | | 94.52 | 97.79 | 97.59 | 98.47 | 98.95 | 98.78 | 98.45 | 98.92 | **99.47** |
| **K::** | | 92.91 | 96.17 | 96.46 | 97.21 | 98.25 | 97.65 | 97.36 | 98.14 | **99** |

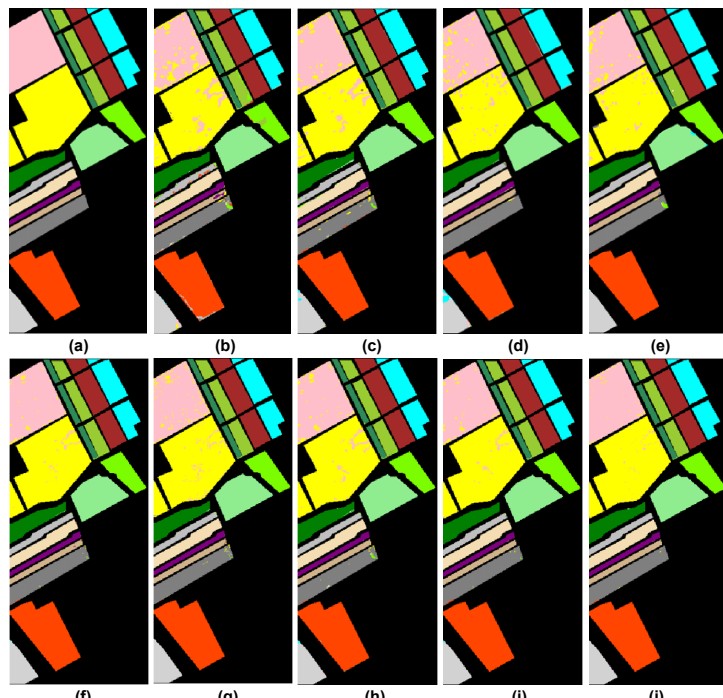

**Figure 13.** (**a**) Ground Truth Image, Classification Maps of (**b**) SVM (**c**) EPF (**d**) SCMK (**e**) R2MK (**f**) ASMGSSK (**g**) MsuperPCA (**h**) 2D-SSA (**i**) 2D-MSSA (**j**) SP-SSA for Salinas dataset.

### 3.4.4. Results from the Houston 2018 Dataset

The quantitative results for the Houston 2018 dataset with 0.2% training samples from each class are presented in Table 5. The corresponding classification map is shown in Figure 14. The best results from the tables are displayed in bold font for comparison. As observed from Table 5, the proposed methods are robust and achieve good classification results even for challenging scenes. The proposed approach improves accuracy from 68.19% to 83.57% for the SVM method. In this case also, the superpixel-based approaches (SCMK, R2MK) display superior performance as compared to non-superpixel based methods (EPF, 2DSSA). Here also, multi-scale window approaches (ASMGSSK, MsuperPCA, and 2D-MSSA) outperform fixed-window based methods as different scales of spatial features are incorporated into the analysis. Figure 14 also highlights the superiority of the proposed method. The salt and pepper noise is reduced by a greater extent, and a smoother classification map is produced with the proposed method.

**Table 5.** Classification results for the Houston Dataset with 0.2% training for SVM, EPF, SCMK, R2MK, ASMGSSK, MsuperPCA, 2D-SSA, 2D-MSSA, and SP-SSA algorithms.

| Class | Samples | SVM | EPF | SCMK | R2MK | ASMGSSK | MsuperPCA | 2DSSA | 2DMSSA | SP-SSA |
|---|---|---|---|---|---|---|---|---|---|---|
| Healthy grass | 9799 | 62.84 | 65.71 | 73 | 74.49 | 81.46 | 76.44 | 76.64 | **84.51** | 79.28 |
| Stressed grass | 32,502 | 84.4 | 85.08 | 83.83 | 86.42 | 89.88 | 88.67 | 83.7 | 90.99 | **91.33** |
| Artificial turf | 684 | **100** | 98.83 | **100** | 99.41 | **100** | **100** | **100** | **100** | **100** |
| Evergreen trees | 13,588 | 83.72 | 73.99 | 80.15 | 84.76 | 87.77 | 82.43 | 80.43 | **91.13** | 87.37 |
| Deciduous trees | 5048 | 43.98 | 36.75 | 50.54 | 51.74 | 63.82 | 55.89 | 46.38 | 58.26 | **73.94** |
| Bare earth | 4516 | 79.32 | 82.67 | 86.28 | 88.46 | 94.01 | 91.1 | 80.64 | 89.04 | **96.21** |
| Water | 266 | 66.42 | 67.8 | 61.89 | 68.06 | 83.08 | 68.18 | 74.62 | 66.92 | **85.71** |
| Residential buildings | 39,762 | 68.26 | 77.29 | 77.52 | 79.2 | 84.37 | 82 | 74.76 | 78.07 | **87.39** |
| Non-residential buildings | 223,684 | 82.57 | 85.51 | 87.3 | 88.31 | 91.64 | 89.18 | 86.05 | 90.63 | **92.73** |
| Roads | 45,810 | 40.41 | 43.7 | 46.79 | 47.33 | 58.86 | 51.39 | 42.75 | 53.25 | **63.89** |
| Sidewalks | 34,002 | 31.36 | 35.3 | 37.58 | 37.51 | 43.74 | 39.25 | 33.65 | **51.35** | 49.71 |
| Crosswalks | 1516 | 4.96 | 9.02 | 8.63 | 9.12 | 13.66 | 9.26 | 7.62 | 5.2 | **14.21** |
| Major thoroughfares | 46,358 | 50.97 | 55.91 | 59.41 | 63.04 | 72.53 | 65.1 | 58.17 | 60.83 | **75.48** |
| Highways | 9849 | 49.7 | 60.35 | 61.19 | 63.93 | 73.83 | 70.78 | 59.55 | 68.92 | **78.56** |
| Railways | 6937 | 79.92 | 85.83 | 92 | 89.52 | 95.29 | 88.07 | 80.22 | 95.84 | **97.29** |

**Table 5.** *Cont.*

| Class | Samples | SVM | EPF | SCMK | R2MK | ASMGSSK | MsuperPCA | 2DSSA | 2DMSSA | SP-SSA |
|---|---|---|---|---|---|---|---|---|---|---|
| Paved parking lots | 11,475 | 54.91 | 63.2 | 74.02 | 69.99 | **84.58** | 74.63 | 64.9 | 74.18 | 84.33 |
| Unpaved parking lots | 149 | 58.78 | 83.11 | 83.67 | 77.03 | 81.63 | 85.71 | 81.08 | 81.76 | **93.06** |
| Cars | 6578 | 43.99 | 47 | 49.33 | 59.66 | 65.09 | 53.78 | 54.51 | 62.85 | **70.22** |
| Trains | 5365 | 40.83 | 40.54 | 58.92 | 51.3 | 77.4 | 61.14 | 52.66 | 79.19 | **79.96** |
| Stadium seats | 6824 | 86.42 | 93.25 | 87.78 | 93.86 | 96.62 | 94.18 | 83.53 | **98.71** | 98.43 |
| **OA::** | | 68.19 | 71.64 | 74.1 | 75.48 | 81.32 | 77.2 | 72.11 | 79.08 | **83.57** |
| **AA::** | | 60.69 | 64.54 | 67.99 | 69.16 | 76.96 | 71.36 | 66.09 | 74.08 | **79.95** |
| **K::** | | 59.13 | 63.41 | 66.52 | 68.28 | 75.82 | 70.55 | 63.94 | 72.61 | **78.73** |

**Figure 14.** (**a**) Ground Truth Image, Classification Maps of (**b**) SVM (**c**) EPF (**d**) SCMK (**e**) R2MK (**f**) ASMGSSK (**g**) MsuperPCA (**h**) 2D-SSA (**i**) 2D-MSSA (**j**) SP-SSA for Houston 2018 dataset.

3.4.5. Statistical Evaluation

The effectiveness of the proposed method was statistically evaluated using McNemar's test. The classification results for all the test cases were compared using this test. The McNemar's test is defined as in Equation (12), where it is assumed that two generic algorithms, named Algorithm 1 and Algorithm 2 are compared.

$$Z = \frac{f_{12} - f_{21}}{\sqrt{f_{12} + f_{21}}} \tag{12}$$

In the equation above, $f_{12}$ indicates the number of samples correctly classified by Algorithm 1 and incorrectly classified by Algorithm 2, and $f_{12}$ indicates the number of samples for the opposite case. The performance of Algorithm 1 is better than Algorithm 2 if $Z > 0$. The differences between Algorithm 1 and Algorithm 2 are statistically significant if $|Z| > 1.96$. In our case, Algorithm 1 is the algorithm proposed in our manuscript, and Algorithm 2 is —sequentially— each one from the list of standard algorithms: SVM, EPF, SCMK, R2MK, ASMGSSK, MsuperPCA, 2DSSA, 2DMSSA.

McNemar's test between the proposed SP-SSA algorithm and the algorithms listed above for the Indian Pines, Pavia University, Salinas, and Houston 2018 datasets are provided in Table 6. The test result clearly reveals that the classification results for the

proposed method were significantly better—in a McNemar's statistical sense—compared with other approaches.

**Table 6.** Statistics of the McNemar Test for the Indian Pines, Pavia University, Salinas, and Houston 2018 datasets.

| Z | Indian Pines | Pavia University | Salinas | Houston 2018 |
|---|---|---|---|---|
| | Proposed Method (SP-SSA) | | | |
| SVM | 42.534 | 56.342 | 37.215 | 61.512 |
| EPF | 39.152 | 48.186 | 25.113 | 57.084 |
| SCMK | 27.467 | 37.428 | 21.421 | 43.115 |
| R2MK | 23.615 | 28.521 | 18.472 | 39.721 |
| ASMGSSK | 10.624 | 16.832 | 8.351 | 18.521 |
| MsuperPCA | 21.524 | 19.441 | 11.486 | 35.431 |
| 2DSSA | 34.321 | 31.084 | 15.321 | 51.322 |
| 2DMSSA | 16.819 | 23.217 | 9.091 | 27.634 |

*3.5. Advantage of Proposed Method over 2D-SSA*

3.5.1. Applying SP-SSA on General Images

In the proposed approach, 2D-SSA is applied on each and every superpixel segmented region. Hence, it can be considered as a local 2D-SSA approach that can extract accurate spatial information on each single object. In the case of global 2D-SSA, features are over-smoothed, and features are not prominent for specific classes. In local 2D-SSA instead, object-specific texture information can be highlighted. In Figure 15, the popular cameraman image and an artificial test image are used to demonstrate the effectiveness of the proposed approach over the 2D-SSA approach.

When the cameraman image is reconstructed using the 2D-SSA method, the Mean Square Error (MSE) comes out to 115.8865; however, when the same image is reconstructed using the proposed SP-SSA approach, the MSE reduces to 93.0468. A similar behavior is also observed with the test image. With the proposed SP-SSA method, the MSE reduces to 237.1038 from 287.5323. This signifies that the proposed method can reconstruct an image with minimum error and can effectively integrate local information during the reconstruction process.

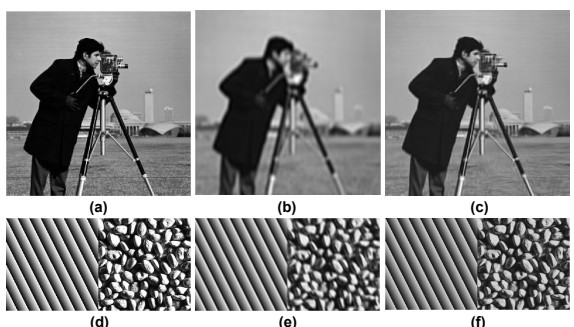

**Figure 15.** (**a**) Cameraman image (**b**) 2D-SSA Reconstructed image [MSE = 115.8865] (**c**) SP-SSA reconstructed image [MSE = 93.0468] (**d**) Test image (**e**) 2D-SSA Reconstructed image [MSE = 287.5323] (**f**) SP-SSA reconstructed image [MSE = 237.1038].

3.5.2. Applying SP-SSA on HSI

The HSI is composed of a stack of 2D images carrying valuable information about each spectral band. To demonstrate the effectiveness of the proposed method, a randomly selected spectral band at 667 nm was considered for our analysis. Figure 16b,c contains the scene as reconstructed by $2D\text{-}SSA$ and $SP\text{-}SSA$, respectively. Since the HSI was acquired over a large area, it includes multiple objects with different textural information. This is a typical case where object-specific reconstruction works better than direct reconstruction.

Textural information can be highlighted accurately by using local reconstruction as opposed to global reconstruction. The error in SP-SSA-based reconstruction is indeed lower as compared to 2D-SSA-based reconstruction. The same conclusion can also be drawn from Figure 16.

In the case of 2D-SSA-based reconstruction, the Mean Square Error (MSE) is 612.4349, while, in the case of SP-SSA-based reconstruction, the MSE is 504.5685. Figure 16d,e contains the difference image for 2D-SSA-based reconstruction and SP-SSA-based methods. It can be clearly observed that edge information is preserved with the proposed method. The SP-SSA-based reconstruction is applied to all spectral bands and generates a modified hypercube with preserved local structure information and minimum noise level. These latter features generally lead to better classification performance.

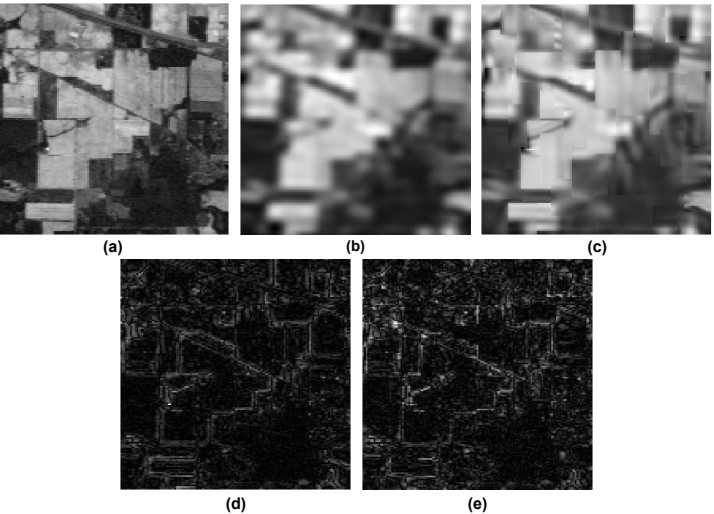

**Figure 16.** (**a**) Original scene at band 667 nm (**b**) Reconstructed scene by 2D-SSA (**c**) Reconstructed scene by SP-SSA (**d**) Difference image for the 2D-SSA reconstructed scene (**e**) Difference image for the SP-SSA reconstructed scene.

## 4. Conclusions and Future Scope

Feature extraction is one of the most crucial steps in HSI classification. It is essential to capture comprehensive spatial and spectral information for accurate feature extraction. For image reconstruction, the conventional 2D-SSA algorithm usually extracts spatial features directly by applying the embedding window to the entire image. However, HSI scenes frequently encompass a broader area and contain several items. As a result, spatial information pertaining to local objects must be recovered. To solve this problem, in the proposed method, a superpixel-based SSA technique was presented, which can capture the object specific spatio-spectral information accurately.

In this work, the original HSI was first divided into various semantic sub-regions by the superpixel segmentation algorithm. Next, each segment was reconstructed individually by applying 2D-SSA. The generated reconstructed HSI was then classified using the SVM classifier, and the final classification map was produced. Local characteristics may be collected effectively in the suggested method since 2D-SSA is applied at the superpixel level. However, two parameters must be adjusted: the amount of superpixels and the embedding window size. Future developments will aim at finding the optimal criteria to determine the parameters of the procedure and to investigate relationships between the characteristics of the HSI and quality of the results.

**Author Contributions:** Conceptualization, P.K.B. and R.P.; methodology, S.S.; software, R.P. and S.S.; validation, S.S., R.P. and P.K.B.; formal analysis, S.S.; investigation, S.S. and R.P.; resources, F.D. and P.K.B.; data curation, S.S.; writing—original draft preparation, S.S.; writing—review and editing, F.D.; visualization, S.S. and F.D.; supervision, F.D.; project administration, F.D. and P.K.B. All authors have read and agreed to the published version of the manuscript.

**Funding:** This research received no external funding.

**Institutional Review Board Statement:** Not applicable.

**Informed Consent Statement:** Not applicable.

**Data Availability Statement:** Publicly available datasets have been utilized for conducting the experiments. Details are provided in the text of the paper.

**Conflicts of Interest:** The authors declare no conflict of interest.

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
