# Peer review of "Superpixel-Based Singular Spectrum Analysis for Effective Spatial-Spectral Feature Extraction"

_applsci, doi:10.3390/app112210876_

Round 1

Reviewer 1 Report

Authors propose a superpixel-based SSA technique in order to capture object specific spatio-spectral information.

The proposed approach seems interesting with respect to accurate classification of HSI. The paper is almost well written. Nevertheless, I have the following comments and recommendations.

Figure 1. would be better transformed to a flowchart.

Maybe adding figures to illustrate how to move from H to C to Z, how Zb is decomposed into Z1b, Z2b,....

It is not clear how to move from the expression of Zb in equation 7 to the one in equation 8.

In subsection 2.2.3:

  • how individual components in (10) are divided into  m subsets S = [S1 , S2 , ..., Sm ] ? add illustrations if possible.
  • what do you mean by an elementary matrix Xi ?
  • how selecting one or more  elementary matrices Xi from each subset is performed exactly ? add illustrations if possible.

In the experiments, how the parameter of Eigen Value Grouping is set ?

Figures 7 and 8 are not readable.

In Tables 2-5, put maxima values in bold.

What about using other classification algorithms ?

Some notations and mathematical formulations need to be checked and corrected.

Check everywhere : Z or Zb

Line 151 : H → Hb

Equation 2 : index i  → b

Line 159 : (r, u) → (ri, ui)

Lines 177, 179 : Q → Qb

Equation 7 is to be checked. Also, give the size of Zb.

Equation 10 is to be checked (index issue)

Equation 11 is to be checked

Reviewer 2 Report

A superpixel-based SSA (SP-SSA) method is proposed in this article. The image is first segmented into multiple regions using a superpixel segmentation approach. Then,  each segment is individually reconstructed using 2D-SSA. The spatial contextual information is preserved, resulting in better classifier performance. Experimental results on four popular benchmark datasets demonstrate the proposed method overperforms the standard SSA technique and various spatio-spectral classification methods. To further express the advantage of the proposed method, it is expected to do more analysis about the experiment results.

Round 2

Reviewer 1 Report

Authors addressed all the comments.